# Polaron in almost ideal molecular Bose-Einstein condensate

O. Hryhorchak[1] and V. Pastukhov[*1]

[1]*Professor Ivan Vakarchuk Department for Theoretical Physics,
Ivan Franko National University of Lviv, 12 Drahomanov Str., Lviv, Ukraine*
(Dated: November 16, 2021)

We discuss properties of a single impurity atom immersed in the spin-1/2 dilute Fermi gas with equal populations of two species in the deep Bose-Einstein condensate (BEC) phase. In this limit, when an almost undepleted BEC of the tightly bound molecules of spin-up and spin-down fermions is formed, we calculate the parameters of an impurity spectrum. It is justified that the leading-order contribution to the impurity energy, while being determined by the two- and three-body scattering processes, is dominated by the former ones.

PACS numbers: 67.85.-d
Keywords: Fermi polaron, Bose polaron, three-body scattering

## I. INTRODUCTION

The last decade has been characterized by a surge of interest to the problem of impurities in either fermionic [1] or bosonic mediums [2], the so-called Fermi and Bose polarons. This is mostly stimulated by the successes of the experimental techniques in the controlled doping of the majority of ultracold Fermi [3, 4] and Bose [5, 6] gases by a single impurity atoms. From the historic perspective, the polarons were an excellent starting point to mimic [7] quasiparticles and their mutual interactions in the condensed matter physics. They also serve as a promising platforms [8] for investigating of the few-body effects, and for describing ferromagnetic phase transitions [9] of the realistic many-body systems in a recent experiments with cold atomic gases.

The generic picture, which survives [10] in two dimensions, of the Fermi polaron behavior typically includes the so-called repulsive [11], attractive [12] and dimer (molecular) branches. The repulsive polaron is metastable while the magnitude of its life-time [13] determines the possibility for observing the Fermi liquid state in the spin-imbalanced system of fermions. Negative couplings lead to the attractive polaronic state [14]. The state-of-art numerical simulations [15–19] generally confirm this phase diagram. The simplest transition occurring in a system when the s-wave scattering length is positive definite and increases, is the molecule-to-polaron one [20–24], which character is a topic of the recent theoretical [25] and experimental [26] debates. A current understanding [26, 27] relies on the first-order transition exactly at absolute zero that is replaced by a smooth crossover behavior at finite temperatures.

A somewhat similar trends are visible in the properties of the low-dimensional Fermi polarons [28]. In 2D, particularly, they possess the repulsive branch [29, 30], the dressed-molecule state [31, 32] when an impurity forms a bound state with a single particle from the Fermi bath.

The further increase of the interaction strength leads to a molecule [33, 34] with finite momentum. The latter two-body finite-momentum bound state can smoothly unbind into the polaronic one at large momenta.

Nonetheless the p-wave trimers can emerge in three [35] and two [36] dimensions within the 'standard' Fermi polaron setup at large mass imbalance, the simplest way to observe the three-body states is to put an impurity in a medium with two macroscopically populated species of fermions. The properties of polarons in the fermionic BCS superfluids were previously discussed in Refs. [37–39]. These studies suggest the smooth crossover from the polaron physics to the trimer impurity states. More recent [40] analysis of impurity immersed in a double Fermi sea constitutes the first-order transition.

Here we address the problem of impurity immersed in the spin-1/2 balanced superfluid Fermi gas in the deep BEC state. This setup, although being the Fermi polaronic, suggests the Bose-polaron-like behavior [41–47] of the impurity that weakly interacts by means of two-, three- and all higher-body induced forces with the host tightly-bound dimers obeying the bosonic statistics. When a width of the dimer bound state is the smallest parameter with the dimension of length in the system, the problem of calculation of the polaron spectrum can be treated by means of perturbation theory with two- and three-body interactions being of the same order magnitude.

## II. FORMULATION

### A. Model

The model under consideration consists of a single impurity atom immersed in the spin-1/2 Fermi particles of equal population $N$ of two species. It is assumed that fermions form a dilute gas of the tight dimers (molecules) in the deep two-body bound states of size $a$ and interact via the short-range potential with impurity. The later potential, in turn, is characterized by the s-wave scattering length $a_i$. Therefore in the following, we adopt

*e-mail: volodyapastukhov@gmail.com

the path integral formulation with the Euclidean action that manifests the underlying physics of the system in the most natural way

$$S = S_f + S_i, \qquad (2.1)$$

where the first term describes two-component fermions [complex Grassmann fields $f_\sigma(x)$] that interact through the bosonic molecular fields $d(x)$

$$S_f = \int dx\, f_\sigma^* \left\{ \partial_\tau - \xi \right\} f_\sigma + g^{-1} \int dx\, d^* d$$
$$- \int dx\, \left\{ d^* f_\downarrow f_\uparrow + \text{c.c.} \right\}, \qquad (2.2)$$

(hereafter we use the summation convention over the spin index $\sigma = \uparrow, \downarrow$); the second one

$$S_i = \int dx\, i^* \left\{ \partial_\tau - \xi_i \right\} i - g_i \int dx\, f_\sigma^* f_\sigma i^* i, \qquad (2.3)$$

is referred to the impurity degrees of freedom (for simplicity, $i^*(x)$ and $i(x)$ are assumed to be spinless Fermi fields) and interaction with the host fermions. Integrations in $S$ are carried out in large $D + 1$ 'volume' $\beta L^D$ and all three fields are periodic with the period $L$ in every of $D$ spatial directions, and anti-periodic with the period $\beta$ in the imaginary-time direction. The couplings $g$ and $g_i$ are assumed to be conventionally rewritten (see, for instance [48]) through the $\uparrow$-fermion–$\downarrow$-fermion and $\sigma$-fermion–impurity vacuum binding energies, respectively. In principle, the ultraviolet divergences in the two-body sectors can be treated by the dimensional regularization as well. We also use shorthand notations $\xi = -\frac{\hbar^2 \nabla^2}{2m} - \mu$, $\xi_i = -\frac{\hbar^2 \nabla^2}{2m_i} - \mu_i$, where $\mu$ and $\mu_i$ are chemical potentials of host fermions and impurity, respectively, moreover $\mu = -\frac{\hbar^2}{2ma^2}$ (dilute $na^D \ll 1$ gas of almost non-interacting molecules) and $\mu_i \propto \frac{\hbar^2}{m_i L^2}$ (in order to ensure the one-particle limit).

## B.  Pure molecules

The ground-state energy of the host spin-up and spin-down fermions with a contact interaction, when $a$ ($a > 0$) is the smallest (except for the range of the two-body forces which is neglected here) parameter with dimension of length in the system, is well understood, almost all fermion pairs macroscopically occupy the $\mathbf{p} = 0$ state in dimensions $D > 1$. At finite temperatures and $D \le 2$, the developed thermal fluctuations completely deplete the BEC of molecules. Therefore in the following, we mainly focus on the $D > 2$ case, where the BEC is robust. By using the standard prescription that anticipate a separation $d(x) = d_0 + \tilde{d}(x)$ of the BEC terms $d_0$ in $S_f$ with the subsequent path-integration over the fermionic fields $f_\sigma$ ($f_\sigma^*$), we end up with the thermodynamic potential

$$\Omega_f^{(0)}/L^D = -|d_0|^2 t^{-1}(2\mu) + \frac{1}{2} g_d |d_0|^4 + \dots, \qquad (2.4)$$

that neglects an impact of the non-Bose-condensed molecules. Here $t^{-1}(2\mu) = g^{-1} + \frac{1}{L^D} \sum_{\mathbf{p}} \frac{1}{2\xi_p}$ is the inverse $\uparrow$-$\downarrow$ fermions two-body $T$-matrix in the center-of-mass frame and $g_d \propto a^{6-D}$ is the molecule-molecule coupling constant which is small and positive-definite. The quantity $d_0$ is not only the (unnormalized) condensate wave function of molecules, but it also equals to the mean-field energy gap in the single-particle excitation spectrum. Neglecting the inter-molecule interaction, making use of the thermodynamic identity $2nL^D = -(\partial \Omega_f / \partial \mu)$, and minimizing $\Omega_f$ with respect to $d_0$, we obtain that $\mu$ up to leading order at small $a^D n$ has to be equal to half of the two-fermion vacuum binding energy, and

$$n = \Gamma(2 - D/2) \left( \frac{m|\mu|}{2\pi\hbar^2} \right)^{D/2} \left| \frac{d_0}{2\mu} \right|^2. \qquad (2.5)$$

Ignoring the effects of the gap, we restrict our further considerations to the limit of extremely dilute gas of host two-component fermions $d_0/|\mu| \propto \sqrt{a^D n} \ll 1$.

## C.  Including impurity

The calculation ideology adopted in the previous subsection can be easily generalized to the system with the mobile impurity. Thus, the program is to integrate out the fermionic fields [with $i(x)$ and $i^*(x)$ included] and keep only condensate terms in the effective action of the non-interacting molecules. So, at the first stage we are dealing with the reduced action $S_{\text{red}}$ that incudes entire $S_i$ and free $\uparrow, \downarrow$-fermions with chemical potentials $\mu$

$$S_{\text{red}} = \int dx\, f_\sigma^* \left\{ \partial_\tau - \xi \right\} f_\sigma + \int dx\, i^* \left\{ \partial_\tau - \xi_i \right\} i$$
$$+ g_i^{-1} \int dx\, d_\sigma^* d_\sigma - \int dx\, \left\{ d_\sigma^* f_\sigma i + \text{c.c.} \right\}. \quad (2.6)$$

For latter convenience, we split the fermion-impurity interactions by introducing an auxiliary bosonic fields $d_{\uparrow(\downarrow)}(x)$ (which are referred below as dimers).

Let us briefly examine Eq. (2.6). It describes the two-component system of non-interacting fermions that, in turn, interact through the complex dimer fields with the impurity fermions. Because it is a single atom the dimer fields $d_\sigma(x)$ gain only one self-energy correction (see Fig. 1) of order unity (all the others have at least a factor $1/L^D$, and therefore, disappear in the thermodynamic limit). In that way, the bosonic propagators $\langle d_{\uparrow(\downarrow)} d_{\uparrow(\downarrow)}^* \rangle$ in a momentum space are found to be equal to

$$-\langle d_{\uparrow(\downarrow),Q} d_{\uparrow(\downarrow),Q}^* \rangle^{-1} = g_i^{-1} + \frac{1}{L^D} \sum_{\mathbf{p}} \frac{1}{\xi_{|\mathbf{p}+\mathbf{q}|} + \xi_i(p) - i\omega_q}$$
$$= t_i^{-1} (i\omega_q - \frac{\hbar^2 q^2}{2M} + \mu + \mu_i), \qquad (2.7)$$

($M = m + m_i$) up to terms of order $1/L^D$. Another obvious observation is related to the sign of the chemical

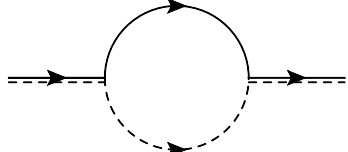

FIG. 1: Simplest diagram contributing to $\langle d_{\sigma,Q} d^*_{\sigma,Q} \rangle$. Solid and dashed lines denote the impurity and fermionic propagators, respectively.

potential $\mu < 0$. Due to the fact that it cannot be drastically changed by a microscopic number of impurities, one readily realizes the absence of the self-energy corrections to the impurity Green's function. Indeed, the simplest diagram in Fig. 2 (a)

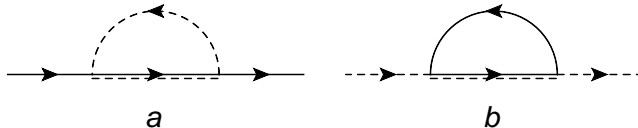

FIG. 2: Diagrammatic representation of impurity (a) and fermionic (b) self energies. Double solid-dashed line stands for propagator (2.7) of bosonic $d_\sigma$ fields.

$$\frac{1}{L^D} \sum_{\mathbf{s}} \int \frac{d\nu_s}{2\pi} \frac{t_i(\mathrm{i}(\nu_s + \nu_p) - \frac{\hbar^2(\mathbf{s}+\mathbf{p})^2}{2M} + \mu)}{\mathrm{i}\nu_s - \xi_s}, \quad (2.8)$$

is non-zero if only the bound state of an impurity and host fermion occurs. Actually, this happens when $|\epsilon_i| > |\mu|$, i.e. in a narrow region $a_i/a = [0, \sqrt{M/m_i}]$ ($a > 0$), where the physics of the system is quite clear. For small $a_i$s (i.e., $a_i \ll a$), the perturbation theory is applicable. This limit for 3D case has been extensively discussed in Ref. [39] even without assumption about a smallness of $a$. The single-particle Green's function of a host fermions [see diagram in Fig. 2 (b)] renormalizes due to a presence of impurity

$$\frac{1}{L^D} \sum_{\mathbf{s}} \int \frac{d\nu_s}{2\pi} \frac{t_i(\mathrm{i}(\nu_s + \nu_p) - \frac{\hbar^2(\mathbf{s}+\mathbf{p})^2}{2M} + \mu)}{\mathrm{i}\nu_s - \xi_i(s)}$$
$$= \frac{1}{L^D} t_i(\mathrm{i}\nu_p - \frac{\hbar^2 p^2}{2M} + \mu) + \mathcal{O}\left(\frac{1}{L^{D+2}}\right), \quad (2.9)$$

where we again set restrictions on $a_i \notin [0, \sqrt{M/m_i}a]$. Now, we are ready to identify the inverse molecular propagator of the $\uparrow, \downarrow$-fermions with the exterior particle immersed. Being calculated at zero momentum and frequency, it determines up to a factor $|d_0|^2$ the density of a grand potential in the simplest approximation [see (2.4)]. The contribution of the self-energy insertions in the Green's functions of individual fermions $f_\sigma$ reads

$$\Omega_f^{(1)} = \frac{|d_0|^2}{2L^D} \sum_{\mathbf{p}} \frac{t_i(2\mu - \frac{\hbar^2 p^2}{2M_r})}{\xi_p^2}, \quad (2.10)$$

where $1/M_r = 1/m + 1/M$ is the atom-dimer reduced mass. Except for the self-energy insertions, there is also a contribution of the same order as $\Omega_f^{(1)}$ to the thermodynamic potential originating from the three-body $f_\uparrow - i - f_\downarrow$ collisions (diagram in Fig. 3). Here the

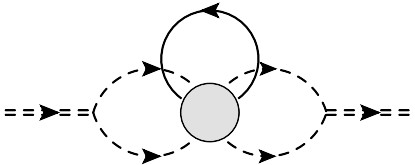

FIG. 3: Three-body contribution to the molecular propagator. With a zero external momenta it equals (up to a constant factor $|d_0|^2$) to the correction to the density of thermodynamic potential. The blob is determined in Fig. 4

blob denotes the three-particle ($\uparrow$-fermion–impurity–$\downarrow$-fermion) vertex function, which in turn can be compactly represented via the two-body ($d_\sigma - f_{\sigma'}$) scattering vertices (see Fig. 4). There are four such vertices $\Gamma_{\sigma\sigma'}$ (here subscripts denote incoming $\sigma$ and outgoing $\sigma'$ dimer lines, respectively). They are grouped in pairs $\Gamma_{\uparrow\downarrow}, \Gamma_{\downarrow\downarrow}$ and $\Gamma_{\downarrow\uparrow}, \Gamma_{\uparrow\uparrow}$, and the fermion-dimer vertices in each pair are mutually connected by means of two linear integral equations. In Fig. 5 the diagrammatic representation of the system of coupled integral equations for functions $\Gamma_{\uparrow\downarrow}, \Gamma_{\downarrow\downarrow}$ is shown. Fortunately, to find out the role of three-body effects in the behavior of impurity in a system of a small size molecules, we only need to know the two-body $d_\sigma - f_{\sigma'}$ on-shell scattering amplitudes. Indeed, while calculating the appropriated correction to the thermodynamic potential (diagram in Fig. 3 with the $D+1$-momenta of the external lines set to zero), we see that impurity bubble gives the factor $1/L^D$ and takes away from the three-body vertex the dependence on its own $D+1$-momentum, and the only contributions are due to fermionic loops from the both sides of the blob. We can now simplify the system of coupled equations in Fig. 5 by going to the center-of-mass frame, performing the frequency integration by encircling the contour in the lower complex half-plane and putting the frequencies of the external lines on the mass-shell $\Gamma_{\uparrow\downarrow}(-P; P|S; -S)|_{\mathrm{i}\nu_p \to \xi_p, \mathrm{i}\nu_s \to \xi_s} = \Gamma_{\uparrow\downarrow}(\mathbf{p}, \mathbf{s})$ [and same for $\Gamma_{\downarrow\downarrow}(-P; P|S; -S)$]. Then these two equations can be equivalently rewritten for linear combinations $\Gamma_{\uparrow\downarrow}(\mathbf{p}, \mathbf{s}) \pm \Gamma_{\downarrow\downarrow}(\mathbf{p}, \mathbf{s})$ of the two on-shell vertices. The expression for the thermodynamic potential contains only sums $\Gamma_{\uparrow\downarrow}(\mathbf{p}, \mathbf{s}) + \Gamma_{\downarrow\downarrow}(\mathbf{p}, \mathbf{s})$ and $\Gamma_{\downarrow\uparrow}(\mathbf{p}, \mathbf{s}) + \Gamma_{\uparrow\uparrow}(\mathbf{p}, \mathbf{s})$ of vertices. Furthermore, the spin-$\uparrow$–spin-$\downarrow$ symmetry arguments constitute that $\Gamma_{\uparrow\downarrow}(\mathbf{p}, \mathbf{s}) + \Gamma_{\downarrow\downarrow}(\mathbf{p}, \mathbf{s}) = \Gamma_{\downarrow\uparrow}(\mathbf{p}, \mathbf{s}) + \Gamma_{\uparrow\uparrow}(\mathbf{p}, \mathbf{s}) = \Gamma(\mathbf{p}, \mathbf{s})$, which leaves us with a simple expression for the three-body correction to $\Omega_f$

$$\Omega_f^{(2)} = \frac{|d_0|^2}{2L^{2D}} \sum_{\mathbf{p},\mathbf{s}} \frac{t_i(2\mu - \frac{\hbar^2 p^2}{2M_r}) t_i(2\mu - \frac{\hbar^2 s^2}{2M_r})}{\xi_p \xi_s} \Gamma(\mathbf{p}, \mathbf{s}) \quad (2.11)$$

The symmetric function $\Gamma(\mathbf{p}, \mathbf{s})$ introduced in $\Omega_f^{(2)}$ satis-

FIG. 4: Three-body vertex written through the $d_\sigma - f_{\sigma'}$ vertices $\Gamma_{\sigma\sigma'}$ (rectangles) and the $d_\sigma$-dimer propagators. Notations are identical to that in Figs. 1,2 ; upper and lower dashed (double dash-solid) lines correspond to $f_\uparrow(d_\uparrow)$ and $f_\downarrow(d_\downarrow)$, respectively.

FIG. 5: The system of two coupled integral equations for vertices $\Gamma_{\uparrow\downarrow}(Q; P|P'; Q')$ and $\Gamma_{\downarrow\downarrow}(Q; P|P'; Q')$.

fies the following integral equation

$$\Gamma(\mathbf{p}, \mathbf{s}) = \frac{1}{\xi_p + \xi_s + \varepsilon_i(|\mathbf{p} + \mathbf{s}|)}$$
$$+ \frac{1}{L^D} \sum_{\mathbf{k}} \frac{t_i(2\mu - \frac{\hbar^2 k^2}{2M_r})}{\xi_p + \xi_k + \varepsilon_i(|\mathbf{p} + \mathbf{k}|)} \Gamma(\mathbf{k}, \mathbf{s}). \quad (2.12)$$

While deriving (2.11), we have explicitly assumed the absence of three-body $(f_\uparrow - i - f_\downarrow)$ bound states. This situation is in contrast to the polaron problem [49] in the bosonic medium, where infinite tower of the Efimov states occurs. The difference from the bosonic case is in a sign in front of the integral term in Eq. (2.12). In order to better understand the peculiarities of the solutions in both bosonic and fermionic cases, let us apply the iterative procedure to Eq. (2.12). At first step one can naively neglect the integral term. Note that the inhomogeneous term behaves like $1/(p^2 + s^2)$, and therefore, by substituting it in the integral in r.h.s., we obtain the correction to $\Gamma(\mathbf{p}, \mathbf{s})$, which large-$p(s)$ tail looks like $\ln p/p^2$ ($\ln s/s^2$). Repeating the iterative procedure, one gets an increasing power of large logarithms in every order. Therefore, the whole series is badly defined in the ultraviolet limit. This is actually the situation when every finite order of perturbation theory (every diagram) for $\Gamma(\mathbf{p}, \mathbf{s})$ is convergent, while the infinite series of diagrams diverges. The main difference between bosonic and fermionic mediums is that in the former case the series has a sign alternation, which weakens the conditions for its convergence and facilitates the numerical treatment of the equation for $\Gamma(\mathbf{p}, \mathbf{s})$.

Without assumption of the molecules formation, there are another types of Efimov trimers in the considered system. They occur in the three-particle system with a non-zero total angular momentum $l$ consisting of a two identical fermions and one impurity. Particularly, in the $l = 1$-channel the Efimov effect is suppressed [50–52] in 3D for all mass ratios below $m/m_i < 13.606...$. A qualitatively similar behavior is expected to be realized in $D \neq 3$ dimensions, where only very light impurities can provide enough attraction between identical fermions to form the Efimov trimers.

Formation of the molecule-impurity bound states in our case is prevented by a very weak but non-zero for all $a \neq 0$ molecule-molecule repulsion. The latter interaction is very important for providing the well-defined thermodynamic limit of the many-body system. Because the absence of any repulsion between tightly-bound molecules would necessary lead to the collapsed BEC state of the system [53], where all molecules form the bound states with an impurity. Therefore, the amount of energy the many-body system of non-interacting bosons gains when a single atom is immersed, is of order of its size $N$.

Finally, we can briefly discuss a question how to modify the above formulation in a case, when the impurity starts to move. The requirement of finiteness of the impurity momentum $\hbar\mathbf{p}_0$ forces us to modify the $i$-propagator by replacing $\xi_i(p)$ with $\xi_i(|\mathbf{p} - \mathbf{p}_0|)$. The Galilean invariance, in turn, provides the possibility equivalently modify the dimer propagators (2.7) by shifting the spatial part of their arguments $\mathbf{q} \rightarrow \mathbf{q} + \mathbf{p}_0$. It turns out, that in order to take into account the impurity motion in a gas of molecules and to calculate the corrections to its energy, we can freely exploit the formulas (2.10) and (2.11) with the shifted dimer propagators. It is worth noticing that, in general, this is not an easy task to be performed, because one must recalculate function $\Gamma(\mathbf{p}, \mathbf{s})$ at finite $\mathbf{p}_0$s first, in order to compute $\Omega_f^{(2)}$. Then, the contributions of $\Omega_f^{(1)}$ and $\Omega_f^{(2)}$ shift the free-particle kinetic energy $\frac{\hbar^2 p_0^2}{2m_i}$ of a moving impurity. At small $p_0$s ($p_0 a \ll 1$), the modified finite-momentum part of the impurity spectrum is parabolic with the effective mass $m_i^*$

$$\frac{m_i}{m_i^*} = 1 + \frac{|d_0|^2}{2L^D} \sum_{\mathbf{p}} \frac{1}{\xi_p^2} \left[ \frac{m}{M} \frac{\partial}{\partial(2\mu)} \right.$$
$$\left. + \frac{m_i}{M} \frac{\hbar^2 p^2}{DM} \frac{\partial^2}{\partial(2\mu)^2} \right] t_i(2\mu - \frac{\hbar^2 p^2}{2M_r}), \quad (2.13)$$

calculated by including $\Omega_f^{(1)}$ only.

## III. RESULTS

For simplicity we carried out the numerical calculations in the limit $a_i \gg a$. Formally, one can think about the unitarity limit $a_i \to \pm\infty$, or gas of a very deeply-bounded molecules. The key physical limitation for positive $a_i$s is the absence of the two-body impurity-fermion $i - f_{\uparrow,\downarrow}$ bound states. Without $a_i$, the remaining free parameters are the impurity-fermion mass ratio $m_i/m$ and spatial dimension $2 < D < 4$. The theorem about small corrections to the thermodynamic potentials identifies the sum $\Omega_f^{(1)} + \Omega_f^{(2)}$ (rewritten through density of molecules) as energy of impurity. Notably both corrections are of the same order magnitude at small $a^D n$. Being written down in the units of a binding energy $2|\mu|$ of molecules, the final expression

$$\frac{\Omega_f^{(1)}}{2|\mu|} + \frac{\Omega_f^{(2)}}{2|\mu|} + \dots$$
$$= a^D n \left[ \Delta e_i^{(1)} + \Delta e_i^{(2)} \right] + o(a^D n), \qquad (3.14)$$

contains two dimensionless functions $\Delta e_i^{(1)}$ and $\Delta e_i^{(2)}$ referring to Eq. (2.10) and Eq. (2.11), respectively. The first one, $\Delta e_i^{(1)}$, can be brought to a simple one-dimensional integral, while the calculations of the second term in (3.14) requires the solution of the integral equation (2.12) at fixed $D$ and $m_i/m$. The results of the numerical calculations (see Fig. 6) reveal the dominance of the two-body impurity-fermion scattering processes over the essentially three-body ones. Such a discrepancy of numerical prefactors near various terms in $\Omega_f$, which are of the same order magnitude in the characteristic small parameter $a^D n$, but originate from the scattering processes with a different number of particles involved, argues that the leading-order contribution to the impurity effective mass is given by Eq. (2.13). Because of a complexity of $\Omega_f^{(2)}$ at finite impurity momentum $\mathbf{p}_0$, we did not manage to confirm the later statement by the direct numerical calculations. It turns out that the correction to the effective mass in (2.13) is positive definite (i.e. the impurity is less agile in the medium) and of order $a^D n$

$$\frac{m_i^*}{m_i} = 1 + a^D n \Delta_i^{(1)} + o(a^D n), \qquad (3.15)$$

with a dimensionless prefactor $\Delta_i^{(1)}$ presented in Fig. 7.

## IV. CONCLUSIONS

Summarizing, we have studied the problem of a mobile impurity immersed in the $D$-dimensional weakly-interacting gas of the tightly bound molecules composed of two fermions with opposite spins. The deep BEC phase leaves the parameters of the impurity spectrum almost unmodified, with the leading-order corrections proportional to $a^D n$ (where $a$ is the width of the two-fermion

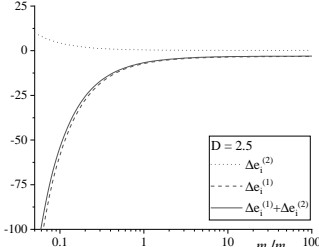
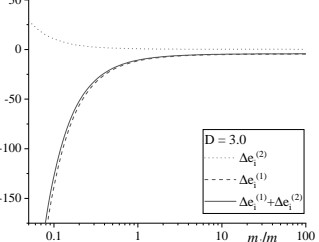
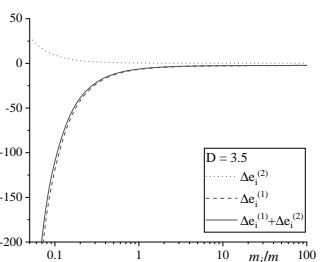

FIG. 6: The dimensionless impurity energy (3.14) as a function of the mass ratio $m_i/m$ in the limit $a_i \gg a$.

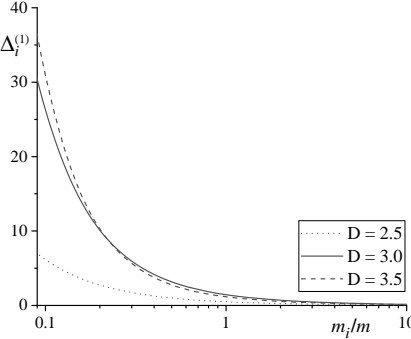

FIG. 7: Numerical prefactor in the leading-order correction ($\propto a^D n$) to the impurity effective mass.

bound state and $n$ is the density of molecules). In general, these corrections originate from the two- or three-body scattering processes involving impurity and one or two fermions, respectively. Our numerical calculations, however, revealed the substantial dominance of the two-

body scatterings over the three-body ones. The later effects are shown to be more distinguishing in a case of the light impurities which inspires hope for an experimental observation of the three-body physics in the molecular BEC polarons.

## Acknowledgements

We are grateful to Dr. I. Pastukhova for careful reading of the manuscript.

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
