# Peer review of "Polaron in almost ideal molecular Bose-Einstein condensate"

_SciPost Physics_

## Round 1 · Referee Report · Anonymous (Referee 1) · 2022-1-17

Report

In this paper, the authors try to study the interaction energy between an impurity and a Fermi superfluid, when the latter is in deep BEC limit. There have been a number of studies in literature on this topic, such as [37-39], and also some few-body calculations that are not cited in this paper:
Phys. Rev. A 90, 041603 (R) (2014); Phys. Rev. A 90, 063614 (2014)

The authors have focused on deep BEC regime of background fermions, and found that the impurity energy is dominated by two-body process. This conclusion seems to be contradictory with those in [38,39], which claim the importance of three-body physics in the same system. In particular, these works show that a logrithmic divergence can appear even in the second-order perturbation energy, which can easily dominate over the mean-field term. The few-body calculations mentioned above also show the importance of three-body physics in modifying the mean-field characterization of impurity energy. Therefore the authors need to explain why their statement is so different from these studies. Before that, I do not think this paper qualified for publication in any journal.

---

## Round 1 · Referee Report · Anonymous (Referee 2) · 2022-1-28

Report

The authors study a single fermionic impurity immersed into a BEC made from fermionic dimers. One could imagine this as the limit of a three-component Fermi gas (red, green, blue), where the density of red fermions is much lower than that of the green and blue ones; the green and blue fermions are in the BEC-regime of a BEC-BCS crossover, and the red fermion acts as a polaron within this superfluid background. This needs to be contrasted to the Fermi polaron problem of a red fermion immersed into a sea of blue ones. Using the diagrammatic T-matrix approach, the authors derive equations for the fermion-, impurity-, and dimer-propagators and vertices in arbitrary spatial dimension D. Observables derived from this are the impurity energy and the impurity effective mass. The authors numerically solve their equations to compute these observables for 2<D<4 and a_i >> a > 0. (Here the fermion-fermion (green-blue) scattering length is a>0, and the fermion-impurity (red-green) scattering length is a_i.)

I think revisions are in order for this manuscript to be publishable.

Let me first comment on the novelty of the research. The problem of a third fermion-component interacting with a superfluid background was first studied by Nishida (Ref 37), from what I can tell for a=a_i. Nishida also discussed the important role of the width of the Feshbach resonance, which does not seem to be discussed in this work at all. (The authors work in the narrow-resonance limit.) In this original work, the distinction between an atom-, dimer, and trimer-phase of the impurity was made.

(1) The authors should clarify in which of these phases "their" impurity or polaron is to be found.

The subsequent works cited in this manuscript, Ref 38 and Ref 39, discuss the properties of this system for other choices of a_i and a, especially including a to be in the BEC regime. Therefore, the authors' claim that the "properties of polarons in the fermionic BCS superfluids were previously discussed in Refs. [37-39]" (first page, right column) does not quite reflect a proper account of previous work. It appears that only the limit a_i >> a > 0 considered by the authors here seems to be new. Therefore, it would be important for the present work to very clearly distinguish the new findings from results obtained earlier in the literature. In particular:

(2) Which of the equations derived are new and are not contained in other references on this topic?

If, for instance, all the equations were know and only the numerical evaluation in Sec. III would be new, then I would find it hard to see why these results necessarily need to be published. Furthermore, the authors of Ref. 38 state at the beginning of Sec V of their work:

"the avoided crossings become broader as fermions are tuned towards the BEC side. As a result, the repulsive atomic branch becomes difficult to identify even for small positive [a_i], which suggests instabilities towards three-body losses".

This seems to suggest that the authors' limit a_i >> a > 0 would fall into this unstable region.

(3) The authors should comment on this stability aspect for positive a_i.

I am also confused by the aspect of dimensionality in this paper. It appears that the authors consider a general dimension D, but eventually only compute for 2<D<4, which basically means D=3.

(4) What statements can be derived in D=2 dimensions from the present analysis? What is the point of plotting the result in a non-physical/non-integer dimension in Fig. 6 and 7?

Furthermore, the presentation of the manuscript needs to be improved to meet the standards of publications in this field:

(5) All quantities used in equations need to be properly defined. For instance, the manuscript is lacking the definition of d_\up, d_\down, \omega_q, \nu_s, \epsilon_i, just to name a few. There is also an inconsistent use of notations, such as \xi and \xi_i being defined without any arguments, but later on these quantities appear with arguments, sometimes as subscripts, sometimes in parentheses, sometimes the arguments are vectors and sometimes they are positive numbers. While the latter aspect probably will not lead to misunderstandings, it should still be avoided.

(6) Below Eq. (2.8) it should be a_i/a "\in" instead of "=" before the interval.

(7) The Figures and Labelling in Figs. 6 and 7 are too small.

The English language will need to be improved in the copy-editing process.

(8) In particular, I believe that below Eq. (2.8) it should be "only if" instead of "if only".

---

## Round 1 · Referee Report · Anonymous (Referee 3) · 2022-1-30

Strengths

1) Discussion of the Fermi polaron problem in the deep BEC limit.

Weaknesses

1) Not adequate discussion of the previous literature on the subject.

2) Not adequate discussion of the validity regime of the obtained results.

3) Section 3 on "Results" not adequately commented.

Report

The authors study the properties of an impurity in a two-component Fermi gas in the deep BEC limit. After discussing a formulation of the problem, they present some numerical results.

I think the subject is interesting, but I cannot support the publication of the paper unless major revisions on the following points are provided:

1) since the problem has been studied and references are quoted, a detailed comparison with previous results, especially the ones in refs. 38-39, is in order. It is not enugh to say "The properties of polarons in the fermionic BCS superfluids were previously discussed in Refs. [37– 39].": the authors should discuss if the approaches produce the same results, if they are compatible, and in case what are the differences.

2) the authors present "for simplicity" results for a_i>>a, but it is not clear if this regime is in the validity range for the approximations presented in the previous section (do the perturbative expansion is reliable? there is stability?). A detailed discussion on this point is necessary.

3) the presentation of results in Section 3 is not adequate in my point of view: instead of varying D between 2 and 4, apparently without significant differences between 2.5 and 3.5, the authors should have focused on D=3, varied a_i/a and compared when possible with available results in literature. The section 3 should be critically increased, also to support in a more convincing way the conclusion about the dominance of 2-body scatterings against 3-body.

4) the presentation of the formulation is instead more clear, but several inaccuracies in it are anyway present: the model 2.2 has some term with hbar=1, but after hbar is present; moreover the connection with the fully fermionic formulation should be presented, and the reason for which the form 2.2 emerges better motivated. The connection with the a_i should be better discussed. The formula 2.6 should be more clearly explained, and the procedure of splitting the fermion-impurity interactions detailed. If the authors do not want to make too long this section, appendices are in my opinion requested.

Without a significant rewriting addressing the previous points, I am afraid to conclude the paper should be not published.

Requested changes

See the points 1)-...-4) of the report.

---

## Editorial Decision

awaiting_resubmission